# Vaccine Literacy and Source of Information about Vaccination among Staff of Nursing Homes: A Cross-Sectional Survey Conducted in Tuscany (Italy)

**DOI:** 10.3390/vaccines10050682

**Published:** 2022-04-25

**Authors:** Chiara Lorini, Francesca Collini, Giacomo Galletti, Francesca Ierardi, Silvia Forni, Claudia Gatteschi, Fabrizio Gemmi, Lorenzo Stacchini, Sophia Papini, Beatrice Velpini, Luigi Roberto Biasio, Guglielmo Bonaccorsi

**Affiliations:** 1Department of Health Science, University of Florence, 50134 Florence, Italy; guglielmo.bonaccorsi@unifi.it; 2Health Literacy Laboratory, Department of Health Science, University of Florence, 50134 Florence, Italy; 3Quality and Equity Unit, Regional Health Agency of Tuscany, 50141 Florence, Italy; francesca.collini@ars.toscana.it (F.C.); giacomo.galletti@ars.toscana.it (G.G.); francesca.ierardi@ars.toscana.it (F.I.); silvia.forni@ars.toscana.it (S.F.); claudia.gatteschi@ars.toscana.it (C.G.); fabrizio.gemmi@ars.toscana.it (F.G.); 4School of Specialization in Public Health, University of Florence, 50121 Florence, Italy; lorenzo.stacchini@unifi.it (L.S.); sophia.papini@unifi.it (S.P.); beatrice.velpini@unifi.it (B.V.); 5Giovanni Lorenzini Foundation, Viale Piave 35, 20129 Milan, Italy; lrbiasio@gmail.com

**Keywords:** vaccine literacy, information, validation, measurement tool, nursing home

## Abstract

Vaccine literacy (VL) mediates the transfer of information and facilitates vaccination acceptance. The aims of this study are to validate the HLVa-IT (Health Literacy Vaccinale degli adulti in Italiano—Vaccine health literacy for adults in Italian language) for the staff of nursing homes (NHs), to measure VL in such a peculiar target group, and to assess its relationship with the sources used to obtain information about vaccines and vaccinations. A survey has been conducted in a sample of Tuscan NHs using an online questionnaire. Eight-hundred and fifty-three questionnaires were analyzed. Two dimensions of the HLVa-IT appeared (functional and interactive/communicative/critical VL). The HLVa-IT interactive/communicative/critical subscale score was slightly higher than the functional subscale, although with no statistical significance. General practitioners (GPs) or other professionals have been reported as the main source of information by most of the respondents (66.1%). The HLVa-IT total score was significantly higher among those who have declared to use official vaccination campaigns (mean score: 3.25 ± 0.49; *p* < 0.001), GPs or other health professionals (3.26 ± 0.47; *p* < 0.001), and search engines (3.27 ± 0.48; *p* = 0.040) as the main sources of information. In conclusion, the HLVa-IT could be reliable test to investigate VL for staff of NHs, and also to highlight criticalities related to information sources.

## 1. Introduction

Vaccination is one of the most relevant interventions to prevent infectious diseases. Despite that, the phenomenon of vaccination refusal has been increasing in the last years: according to estimates by the World Health Organization (WHO), vaccination saves 2 to 3 million lives worldwide every year but, nevertheless, a considerable number of children and adults are not getting vaccinated, leading to outbreaks of vaccine-preventable diseases and avoidable deaths [1]. The Coronavirus Disease 19 (COVID-19) pandemic has strongly highlighted the impact of vaccine concerns by part of the general population: vaccine delay or a manifest refusal of vaccination, i.e., vaccine hesitancy, have been seen despite the availability of vaccines and vaccination services [2]. In particular, in the current pandemic, misinformation on the disease and on vaccine safety, as well as the conflicting communications from experts and the media, have generated ups and downs in COVID-19 vaccination acceptance, also as a consequence of the ups and downs of viral waves [3]. Vaccine hesitancy is becoming an important challenge for public health, although it was already well-known by experts in the pre-pandemic period and can represent a relevant issue even in the future if not addressed in an effective way. In Italy, vaccine hesitancy is often related to the spreading of fake news on diseases and related vaccinations [4]. Misinformation and distrust in the authorities, government, or scientists have also led to an increase of the anti-vaccination movement, with an escalation of reactions following changes in vaccination policy, such as mandatory childhood vaccinations against 12 diseases in 2017 or mandatory COVID-19 vaccination for many work categories [5,6,7,8].

Vaccine literacy (VL) could be one of the key elements to combat vaccine hesitancy. In line with the concept of health literacy, it aims to ensure that everyone understands what they need to know and do to get vaccinated and occurs when the skills and abilities of people align with the content, processes, and systems needed to access vaccines and get vaccinated [9]. Moreover, VL can be considered as a tool mediating the transfer of information and facilitating vaccination acceptance, so communication about vaccination should be lined up with peoples’ VL [10]. This is particularly important as the information environment on health and vaccination has been increasingly complex in the era of the COVID-19 pandemic: digital sources and social media have been added to traditional media and to health professionals, and in some cases, have completely replaced them. Moreover, conspiracy beliefs and misinformation on vaccinations are widely disseminated on social media, both in the pre-pandemic and in the pandemic period [11,12,13,14,15,16].

Before the COVID-19 pandemic, very few tools had been implemented to assess VL [10,17,18], while from 2020 onwards, specific tools devoted to measure COVID-19 VL have been developed and used [19,20,21,22,23]. Recently, Biasio et al. developed the HLVa-IT (Health Literacy Vaccinale degli adulti in Italiano—Vaccine health literacy for adults in Italian language) to measure VL among Italian adults [17].

A healthcare context where vaccine hesitancy could lead to worse outcomes is that of nursing homes (NHs). Elderly people living in NHs present many comorbidities and a general health status that can rapidly and strongly get worse if associated with infectious diseases. Moreover, those who live in NHs represent a priority population for various vaccinations, such as those against seasonal influenza, COVID-19, and pneumococcal disease [24,25,26]. For these reasons, vaccination of the entire staff of NHs is fundamental to contrast the spread of infectious agents, and particularly those responsible for respiratory diseases: vaccination can in fact serve both to limit absenteeism—which can provoke difficulties in providing appropriate assistance—and to guarantee residents’ health. Despite that, vaccine hesitancy and vaccine concerns among staff of NHs are well-documented [27,28,29,30,31]. In particular, two studies conducted in Tuscany (Italy) in 2018 and 2020 revealed low influenza vaccination uptake among the staff of NHs (lower than 25%), concern about the uselessness of the influenza vaccine, low health literacy levels, as well as lacking or wrong information about influenza disease and vaccination [30,31].

To the best of our knowledge, to date, no studies with the purpose of measuring VL among the staff of NHs have been published. The aims of this study, as part of a larger research project, are to validate the HLVa-IT for the staff of NHs, to measure VL in such a peculiar target group, and to assess its relationship with the sources used to obtain information about vaccinations.

## 2. Materials and Methods

The study adopted a cross-sectional design and was conducted according to the principles of the Helsinki Declaration. It was proposed by email to the Chief Officers of each Tuscan NH (about 300) by the Regional Health Agency of Tuscany, and 98 of them voluntarily joined. In each NH, all the staff members were included, regardless of the type of employment contract, the job responsibilities, and the qualifications.

Data were collected using two online questionnaires. The first one was filled in by each Chief Officer to collect general information on each NH. The second one was filled in by the staff members to collect individual data regarding demographic, educational, and health information, attitudes, and beliefs about vaccination, influenza vaccination, as well as VL (using the HLVa-IT, see below) and sources of information on vaccination (see below). The Chief Officers who joined the study sent the link of the online questionnaire to the staff members of his/her NHs, inviting them to fill it in. Nonetheless, the Chief Officers did not have access to collected questionnaires and did not know who filled them in. On the other hand, filling in the questionnaire was voluntary, and identification by the research group of those who filled in the questionnaire was not possible.

For the purposes of this study, only data collected using the second one (i.e., the questionnaire filled in by the staff members) were considered.

The survey was conducted in July–August 2020.

Considering the results of Biasio’s study [18] as a reference for power calculation (HLVa-IT: mean score 3, standard deviation 1.1), taking into account 0.2 as the score difference to be detected, a sample size of 700 led to a power estimation higher than 99%.

### 2.1. HLVa-IT 

The HLVa-IT is a self-rated measure of VL in adults [18]. It has been built on the so-called Ishikawa test, that is a self-administered questionnaire specific for diabetic patients which includes three health literacy scales [32]. It is composed by 14 Likert-type items aimed at assessing functional VL (items 1–5), interactive/communicative VL (items 6–10), and critical VL (items 11–14), according to Nutbeam’s definition of health literacy domains [33]. Each answer contains four possible choices, with an associated score (for the functional items: 4—never, 3—rarely, 2—sometimes, 1—often; for the interactive and the critical items: 1—never, 2—rarely, 3—sometimes, 4—often); the higher the score, the higher the VL.

Figure 1 reports the complete HLVa-IT, translated into English language.

In the validation study, interactive and communicative items appear to belong to the same domain, while the functional items converge into another specific domain.

The HLVa-IT contains two filter questions. In particular, both before submitting the functional items and before the interactive/communicative and critical items, a filter question is included, in order to select the subjects who have had previous experience with written documents on vaccines and vaccinations (“have you ever read vaccine materials, such as leaflets or posters in doctors’ or public health units’ offices, recommending vaccinations?”) and those who have ever thought or have been advised to vaccinate themselves (“have you ever thought or been advised to vaccinate yourself against one or more diseases?”).

In the validation study, the HLVa-IT had shown good face and construct validity detected by a sample of experts in the fields of vaccines and public health, and good psychometric characteristics were confirmed during the pilot study conducted in a sample of adults recruited in the waiting room of clinical or administrative offices. Moreover, the score presented a significant correlation with knowledge on vaccines and vaccinations, as well as with vaccine acceptance [18].

The scale and subscale scores were calculated as the mean value of all the items or of the items included in each subscale, as emerging from the Principal Component Analysis (PCA) and the assessment of the internal consistency (see Section 2.3). For scoring calculation, only questionnaires without missing responses in all the items of the HLVa-IT were considered.

### 2.2. Sources of Information 

The main source of information regarding vaccination was investigated using a closed question which allowed only one answer. The response options were the following: leaflets and brochures; posters hanging in waiting rooms of clinical or administrative offices; television or radio; vaccination campaigns conducted by public bodies; general practitioner or other healthcare professionals; friends, family members, or social networks; social media; search engines.

### 2.3. Statistical Analysis

Data were presented as mean and standard deviation or median and interquartile range. Normality was assessed using the Kolmogorov–Smirnov test.

A PCA was conducted to examine the factor structure of the questionnaire. Varimax rotation was used, and missing data were deleted. To establish the number of principal components (PCs), we used Kaiser’s rule (Kaiser–Guttman criterion). According to the rule, the PCs were retained if their eigenvalues were greater than or equal to 1. Then, the most contributing variables for each dimension were identified. Internal consistency of HLVa-IT and its subscales were assessed by Cronbach’s alpha and the Omega total coefficient.

Finally, the association between the HLVa-IT score and the use of each source of information was assessed using the Student’s *t*-test or Mann–Whitney U test, where appropriate. Student’s *t*-test was also used when the sample size in each subgroup was above 30, regardless of the results of the Kolmogorov–Smirnov test. Bonferroni correction was used in order to avoid false-positive interference due to multiple comparisons.

Univariate logistic regression analysis was conducted to evaluate the role of HLVa-IT (total score and subscale scores) in predicting the source of information, adjusting for educational level, profession, and language. In particular, for each source of information, three different models were performed depending on the predictor included, and according to the results of the PCA: for model 1, HLVa-IT total score, for model 2, HLVa-IT functional subscale score, and for model 3, HLVa-IT interactive/communicative/critical subscale score. Results were expressed as the Odds Ratio (OR) of using the source of information with respect to not using it.

The analyses were conducted using R version 4.1.2 (Bird Hippie) considering 0.05 as the alpha level.

## 3. Results

### 3.1. Sample Description

In total, 1794 questionnaires were collected. After excluding those workers who had not had previous experience with written documents on vaccines and vaccinations and those who had ever thought or had been advised to vaccinate themselves (filter questions), 858 questionnaires were analyzed for the aims of this study. The characteristics of the sample are reported in Table 1. Staff members were most frequently females (86.7%), with Italian as their mother language (84.7%), with a high school degree (46.6%), and working as assistants or aids (58.6%). Their median age was 45 years old (IQR: 33–52; mean: 43.7 ± 11.2). Excluded questionnaires (*n* = 936, 53%) were filled in by workers significantly older than those included in the analyses (mean age: 45.1 ± 10.8 vs. 43.7 ± 11.2), less frequently graduated (17.3% vs. 26.8%), and less frequently healthcare workers (25.8% vs. 31%).

### 3.2. HLVa-IT: Principal Component Analysis and Internal Consistency

Table 2 describes the results of the PCA with varimax rotation. Three dimensions emerged (eigenvalues greater than or equal to 1), that explained 64.78% of the variance. Considering the contribution of each item in the dimensions (Figure 2), it appeared that two dimensions included all the items well. In particular, the items devoted to measuring functional VL (items 1–5) are included in the second component, while those of interactive/communicative VL (items 6–10) and critical VL (items 11–14) are included in the first component. For these reasons, we decided to consider two dimensions—not three—for all subsequent analyses, although with a reduction of the explained variance (56.75%).

Either the entire scale or the two subscales presented good internal consistency, with Cronbach’s alpha ranging from 0.85 to 0.89 and the Omega total coefficient from 0.90 to 0.92 (Table 3).

### 3.3. HLVa-IT: Item Responses and Score Description

In Table 4, item responses are described. For all the items, less than 9% of missing responses were observed. Moreover, for all the items, a high percentage of respondents reported the lowest value (31–46.5% of “often” for the HLVa-IT functional subscale items; 30.5–54.1% of “never” for the HLVa-IT interactive/communicative/critical subscale items).

The HLVa-IT total score as well as the subscale scores were not normally distributed. The mean and median values of the HLVa-IT total score were 3.19 (SD: 0.49) and 3.21 (IQR: 2.82–3.59), respectively. The HLVa-IT functional (HLVa-IT-F) subscale score distribution was similar to that of the total scale score, with a mean value of 3.17 (SD: 0.69) and a median of 3.20 (IQR: 2.80–3.80). The HLVa-IT interactive/communicative/critical (HLVa-IT-ICC) subscale score was slightly higher than the functional subscale, although with no statistical significance (*p* = 0.22), with a mean value of 3.21 (SD: 0.59) and a median of 3.33 (IQR: 2.89–3.67).

### 3.4. HLVa-IT and Main Sources of Information

GPs or other professionals have been reported as the main sources of information regarding vaccines and vaccinations by most of the respondents (66.1%), followed by official vaccination campaigns (56.1%); on the contrary, social media has been reported as the main source of information by a minority (8.2%) (Figure 3).

Appendix A report on the distributions (mean values and error bars) of the HLVa-IT scores with respect to the main sources of information on vaccine and vaccination used by the staff members of the NHs.

As for the HLVa-IT-F subscale score, no statistically significant differences emerged according to using or not using each source of information. On the other hand, the HLVa-IT-ICC subscale score was significantly higher among those who declared to use official vaccination campaigns (mean score: 3.28 ± 0.58; *p* < 0.001), GP or other health professionals (mean score: 3.32 ± 0.51; *p* < 0.001), and search engines (mean score: 3.32 ± 0.54; *p* < 0.001) as the main sources of information; on the contrary, it was significantly lower among those who declared to use social media (mean score: 2.91 ± 0.68; *p* = 0.01) as the main source of information. Finally, considering the HLVa-IT total score, it was significantly higher among those who declared to use official vaccination campaigns (mean score: 3.25 ± 0.49; *p* < 0.01), GP or other health professionals (3.26 ± 0.47; *p* < 0.001), and search engines (3.27 ± 0.48; *p* = 0.04) as the main sources of information.

Table 5 reports the results of the univariate logistic regression analyses to assess the role of HLVa-IT (total score, functional subscale score, and interactive/communicative/critical subscale scores) in predicting the main sources of information regarding vaccines and vaccinations. After adjusting for educational level, profession, and language, the HLVa-IT total score emerged as a positive predictor of the use of medical or official vaccination campaigns (OR = 1.47), GP or other health professionals (OR = 2.37), and search engines (OR = 1.65), and as a negative predictor of the use of social media (OR = 0.48). As for the HLVa-IT-F subscale score, it emerged as a negative predictor of the use of medical or official posters (OR = 0.76) or TV/Radio (OR = 0.75) as sources of information. Finally, the HLVa-IT-ICC subscale score was a positive predictor of the use of official vaccination campaigns (OR = 1.49), GP or other health professionals (2.68), or search engines (OR = 1.63), and it was also a negative predictor of the use of social media (OR = 0.46).

## 4. Discussion

The aims of this study were to validate the HLVa-IT, to measure VL, and to assess its relationship with the sources used to obtain information about vaccinations in the staff of NHs.

A PCA was conducted in order to investigate if fewer dimensions could underpin the 14 items distributed across the 3 HLVa-IT areas and reduce the redundancy of information. Although three dimensions were identified according to the eigenvalues, the contribution of each item in the dimensions led us to consider only two dimensions in the final results. This confirms what was described by Biasio et al. [18], although with a lower explained variance (62.7% vs. 56.8%). The assessment of the internal consistency confirmed the validity of the scale and of the two subscales (identified by the two dimensions): Cronbach’s alpha and the total Omega coefficient showed a good internal consistency, either for the entire scale or for the two subscales. Our results appear aligned with those reported by Biasio, confirming the robustness of the tool in measuring VL even in different contexts; in fact, even if both populations—the original one and that of the NH staff—were mainly formed by women, Biasio’s study investigated a general population of 200 individuals with a mean age of 63 years old, different to those involved in our study (858 NH professionals with a mean age of 45 years old). Moreover, while in our study the questionnaire was administered online, Biasio’s study was conducted by the PAPI (paper-and-pencil) method. These results show that the HLVa-IT can also be considered valid for staff of NHs using a different administration mode.

Moreover, for all the items, less than 9% of missing responses were observed (range: 6.87–8.86%). Missing responses are good indicators of item difficulty/comprehensibility, as well as of acceptability by the respondents; from this perspective, our data are a further confirmation of the validity of the HLVa-IT. On the other hand, 53% of the staff members who joined the study reported to neither have ever read vaccine materials, such as leaflets or posters in doctors’ or public health units’ offices, recommending vaccinations, nor to have ever thought or been advised to vaccinate themselves against one or more diseases (filter questions), and thus were excluded from the assessment of VL. The high size of this subgroup led to the need of rethinking filter questions, in order to better investigate VL using the HLVa-IT. In fact, excluded and included questionnaires were filled in by staff members with different characteristics: the former were older, with lower educational levels, and were less frequently employed as healthcare professionals than the latter. Since the relationship between these variables and VL has been rarely investigated and without consistent results [20,34,35], it is not clear whether having excluded those questionnaires has impacted the estimation of VL levels.

The mean HLVa-IT total score was 3.19 (from a maximum of 4). The values obtained for each subscale differed slightly, by 0.04 points, with the HLVa-IT-ICC subscale score showing a mean score of 3.21, while for the HLVa-IT-F subscale, a mean score of 3.17 was found. The values are similar to those reported by Biasio in a different population [18]: staff of NHs seem to present VL levels similar to those of the general population recruited in the waiting rooms of general practitioners, highlighting that healthcare skills and competences—either acquired through specific training courses or as a consequence of working in NHs—seemingly do not influence VL.

A survey conducted by Aharon [36], adopting a similarly structured questionnaire, administered to parents of children younger than 9 years old, reported scores comparable to those of our research: 3.0 for the functional and interactive/critical subscales, and 3.3 for the interactive/critical one. The minimal differences among the three surveys can be hypothetically attributed to the fact that younger people are more likely to actively research and process information, allowing us to hypothesize a particular role of age in explaining similar values of items belonging to the interactive/critical subscale.

In our study, a specific section of the questionnaire investigated which sources of information were most consulted by respondents to obtain information on vaccines and vaccinations. Overall, GP or other health professionals were indicated by 66% of respondents, followed by official vaccination campaigns (56%) and by medical or official posters (48%). This was an expected result since the population is formed by workers of a healthcare setting, caring for elderly people. Conversely, the use of social media was less frequent (8.2%). Therefore, NH staff mainly rely on expert sources of information, such as healthcare professionals with whom they can interact and ask for further information, or tools (such as posters or campaigns) produced by the healthcare system, rather than those coming from social media, that can appear more superficial, chaotic, and contradictory, as well as instigators of conspiracy hypotheses. This leads us to hypothesize that social and healthcare professionals have a greater ability than the general population to move across the health information landscape, as well as a higher level of trust in the healthcare system information channels. This appears to be quite dissimilar with respect to the results reported by Costantini [21] in a study involving Japanese familial caregivers, in which national television was the most popular source of information on COVID-19 for the youngest caregivers (56%), followed by social media (37%), doctors (35%), national newspapers (29%), local television (27%), family (24%), and local newspapers (20%).

As for the HLVa-IT-F subscale score, no statistically significant differences emerged according to using or not using each source of information. On the other hand, the HLVa-IT-ICC subscale score was significantly higher among those who declared to use official vaccination campaigns (mean score: 3.28 ± 0.58; *p* < 0.001), GP or other health professionals (mean score: 3.32 ± 0.51; *p* < 0.001), and search engines (mean score: 3.32 ± 0.54; *p* < 0.001) as the main sources of information; on the contrary, it was significantly lower among those who declared to use social media (mean score: 2.91 ± 0.68; *p* = 0.010) as the main source of information. Finally, considering the HLVa-IT total score, it was significantly higher among those who declared to use official vaccination campaigns (mean score: 3.25 ± 0.49; *p* < 0.001), GP or other health professionals (3.26 ± 0.47; *p* < 0.001), and search engines (3.27 ± 0.48; *p* = 0.040) as the main sources of information.

Coming back to our results and observing them through the lens of the two subscales, the score of the HLVa-IT-ICC subscale was significantly higher among those who declared that they used the official vaccination campaigns, GPs, or other health professionals as their main sources (although in this case the difference with the level of the subscale of those who did not indicate the same sources was minimal), and in particular for search engines. Social media, on the other hand, was mostly reported by respondents that showed lower scores on the HLVa-IT-ICC subscale, making room for the hypothesis of a difference in attitudes towards online information according to the individual level of interactive/communicative VL. The results were confirmed by the logistic regression analyses, after controlling for educational level, profession, and language. Although causal relationships cannot be assessed in this study due to the cross-sectional design, we can argue that those who show a higher value on the HLVa-IT-ICC subscale appear to be more likely to seek information online in a more proactive way through search engines; on the other hand, those who show lower levels of the subscale seem to refer passively to online information available through social networks, with a less critical attitude in “weighing” information. This supposed relationship needs to be further investigated with targeted and sensitive tools.

In the logistic regression analysis, TV and radio, as well as medical or official posters, were negatively predicted by the score on the HLVa-IT-F subscale—the higher the score, the lower the likelihood of using those as main sources of information for vaccines and vaccinations, probably due to lower interest in an active search for specific information. These results confirm the need of developing a close collaboration with journalists and the media [37], so as to make the information on vaccine and vaccination available on TV and radio as understandable as possible by people with low functional VL.

The use of social media as a source of information does not seem to be a negative factor in itself, as emerged from a study carried out in the USA and UK in 2021 [11]. In this survey, which involved US and British citizens, a negative relationship emerged between intentions to get vaccines and consulting social media, but only if the latter were used as a substitution of traditional media. In this circumstance, therefore, social media appears as an inadequate substitute when reflecting a passive acceptance of contents. From this perspective, critical VL—as well as critical general health literacy—needs to be reinforced so as to increase the individuals’ ability to evaluate the different topics provided within the information scenario [38]. This has been particularly important during the COVID-19 pandemic: in order to browse the large amount of information—true and fake—easily accessible to anyone with an internet connection, individuals have to apply critical health literacy to the content evaluation [38].

Rowlands and Nutbeam in 2013 introduced the concept of “inverse information law” [39]. In their paper, they stated that health literacy skills are not only dependent on cognitive abilities, but also on exposure to different forms of communication and message content. From this perspective, we can argue that people with lower VL skills tend to mainly use the information sources that offer the simpler message (short texts, with pictures and videos, without extensive explanations), such as social media, posters, TV, and radio. Similar considerations emerged from a recent study on health literacy [40] conducted in Hong Kong. The study highlights the matching of high levels of problematic or inadequate health literacy with low VL. For example, the participants’ low capability to find out relevant information about the vaccine, and to interpret and evaluate it, results in an increased difficulty in making vaccination decisions. These results suggest the need to develop a more tailored communication using those sources of information most popular among people with low vaccine literacy, so as to adequately improve their general and specific skills and make them able to understand even more complex messages and to consult an increasingly wide range of information sources.

This study had some limitations, mainly due to the fact that the questionnaire was administered to the workers of the organizations who voluntarily joined the survey, allowing the possibility of a selection bias in engaging those NHs that show a greater sensitivity or propensity towards vaccination issues. In addition, the information sources surveyed were affected to some extent by the influence of the pandemic context, which may have affected the dynamics of consultation. Furthermore, the high number of excluded questionnaires due to the negative responses to the filter questions may have led to biases in the estimation of VL levels.

Finally, many practical implications of our results can be listed. First, despite the limitation linked to the need of rethinking the application of the filter questions, the results suggest the validity of the HLVa-IT in measuring VL among the staff of NHs. This result gives the premise of using this tool to study VL in such a peculiar worker group, in order to describe subgroups with lower VL, to monitor the level over time, and to evaluate the effectiveness of interventions aimed to increase VL. Second, the relationship between specific VL skills—functional and interactive/communicative/critical competences—and the main sources of information regarding vaccines and vaccinations paves the way towards tailored communication strategies.

In view of future developments, it would be interesting to explore how HLVa-IT can be applied targeting both general and specific populations engaged in the infodemic context of COVID-19 vaccination. In fact, some of the questions of the HLVa-IT have already been used in an online survey aimed at assessing knowledge and attitudes toward COVID-19 vaccines in the early stages of development [19,20].

## 5. Conclusions

The results of our study suggested that HLVa-IT could be a reliable test to investigate VL—either as a multidimensional construct or in its dimensions of functional or interactive/communicative skills—for different populations with respect to those included in the validation study and, specifically, for staff of NHs. The association between VL and some sources of information revealed the need to invest in promoting specific components of VL by using tailored interventions, as well as in enhancing digital and media skills. In particular, the increase of NH staff’s capacity of recognition, analysis, and evaluation of the arguments in order to make appropriate health choices has to be recognized as a public health priority, especially in the context of COVID-19.

## Figures and Tables

**Figure 1 vaccines-10-00682-f001:**
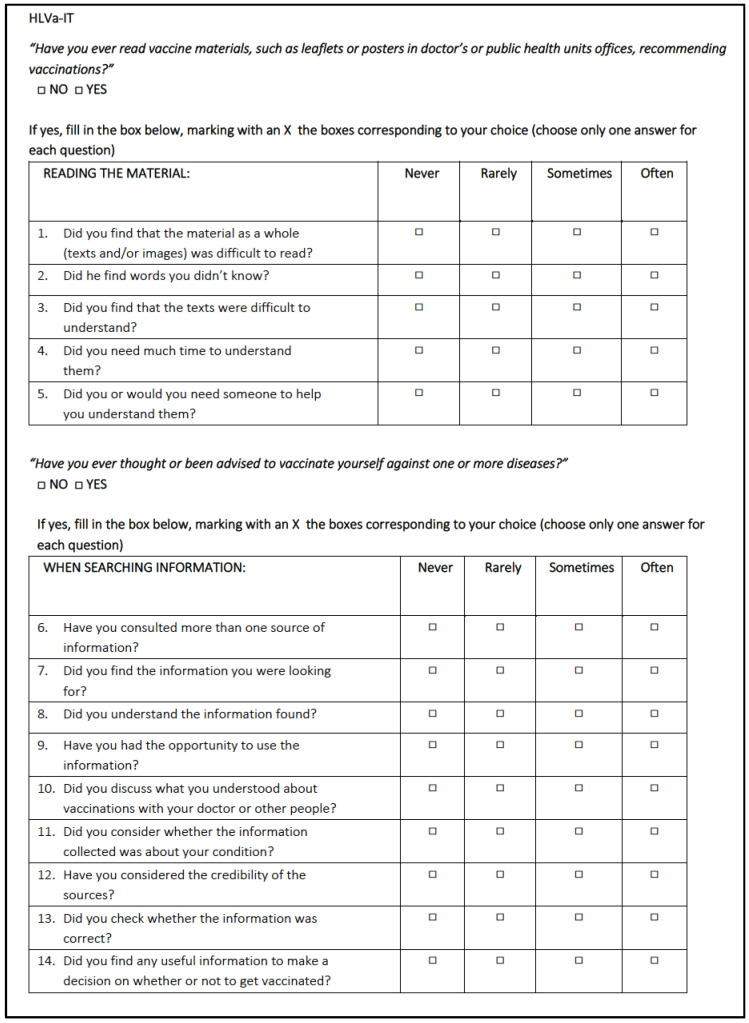
The HLVa-IT included in the study.

**Figure 2 vaccines-10-00682-f002:**
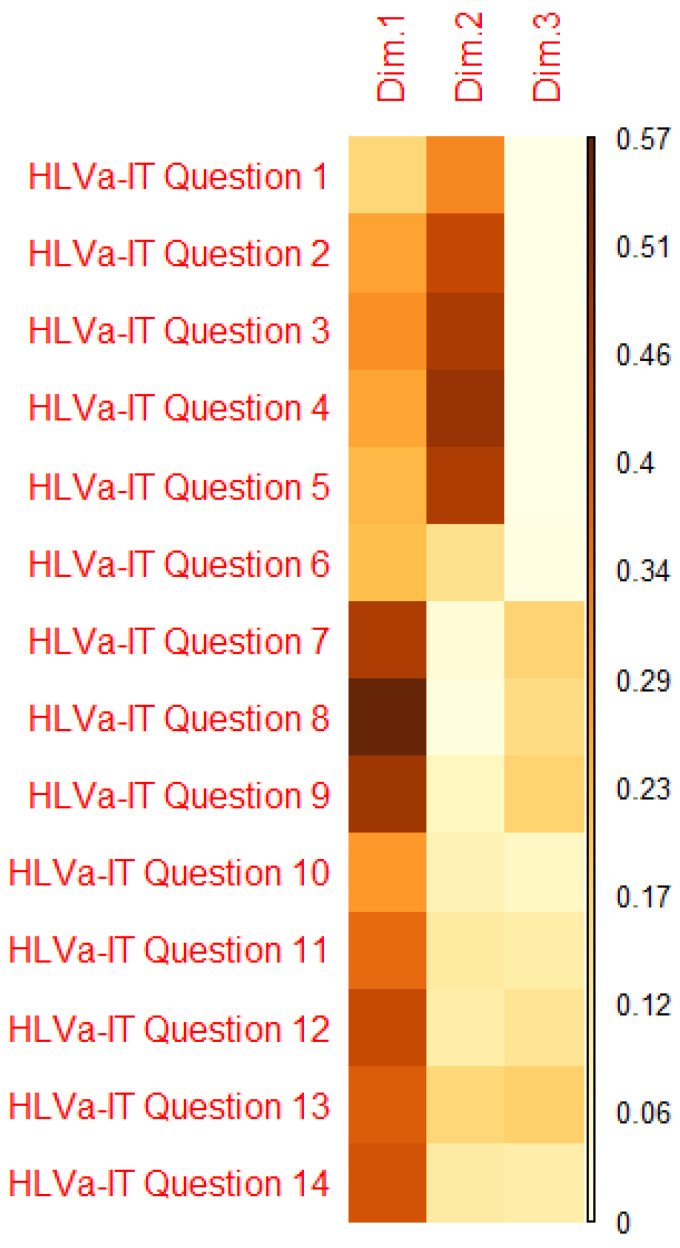
Quality of representation of variables on the factor map.

**Figure 3 vaccines-10-00682-f003:**
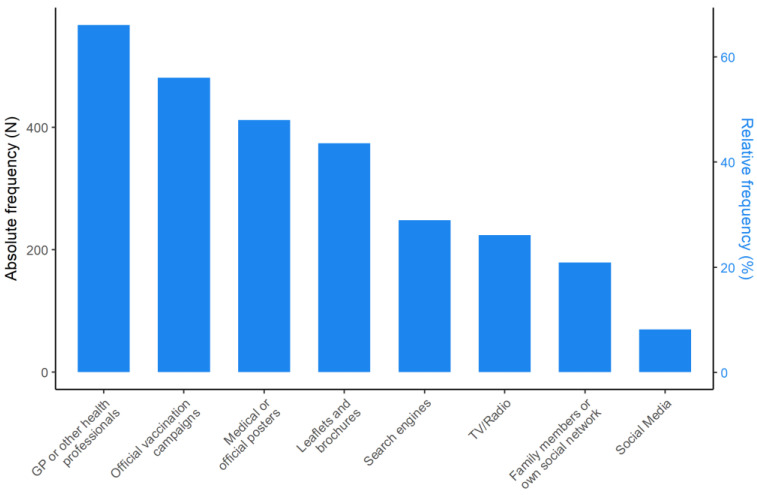
Description of the main sources of information regarding vaccines and vaccinations used by the staff members of NHs.

**Table 1 vaccines-10-00682-t001:** Characteristics of the sample (*n* = 858).

Variables		*n*	%
Sex	Female	744	87.0
	Male	106	12.0
	NA *	8	1.0
Language	Italian	727	84.7
	Others	108	12.6
	NA *	23	2.7
Educational level	Less than high school diploma	220	25.6
	High school degree	400	46.6
	Bachelor’s degree and higher	230	26.8
	NA *	8	1.0
Profession	Nurses	139	16.2
	Assistants/aides	503	58.6
	Physiotherapists	41	4.9
	Health educators	34	4.0
	Other clinical staff	52	6.1
	Cleaning staff	34	4.0
	Other nonclinical staff	43	5.0
	NA *	12	1.4

* NA = not available (missing data).

**Table 2 vaccines-10-00682-t002:** Principal Component Analysis (PCA): eigenvalue, percentage of explained variances of a single component, and cumulative variance percent.

Component	Eigenvalue	Percentage of Explained Variance	Cumulative Variance Percent
Item 1	4.88	34.87	34.87
Item 2	3.06	21.88	56.75
Item 3	1.12	8.03	64.78
Item 4	0.80	5.72	70.50
Item 5	0.66	4.73	75.24
Item 6	0.63	4.51	79.75
Item 7	0.52	3.75	83.49
Item 8	0.45	3.24	86.73
Item 9	0.40	2.86	89.59
Item 10	0.36	2.61	92.20
Item 11	0.34	2.46	94.66
Item 12	0.30	2.11	96.77
Item 13	0.26	1.87	98.64
Item 14	0.19	1.36	100.00

**Table 3 vaccines-10-00682-t003:** Internal consistency: standardized Cronbach’s alpha and Omega total coefficient.

Scale and Subscales	Standardized Cronbach’s Alpha	Omega Total Coefficient
HLVa-IT	0.85	0.91
HLVa-IT Functional Component	0.89	0.92
HLVa-IT Interactive/Communicative and Critical Component	0.87	0.90

**Table 4 vaccines-10-00682-t004:** HLVa-IT: item responses.

Subscale	Items	Response Options *n* (%)					Mean ± SD	Median (IQR)
		1—often	2—sometimes	3—rarely	4—never	*Missing*		
**Functional subscale** “*Have you ever read vaccine materials, such as leaflets or posters in doctors’ or public health units’ offices, recommending vaccinations?*” *If yes…*	1. Did you find that the material as a whole (texts and/or images) was difficult to read?	39 (4.55)	208 (24.24)	271 (31.59)	281 (32.75)	59 (6.87)	2.99 ± 0.90	3 (2–4)
	2. Did you find words you didn’t know?	27 (3.15)	206 (24.01)	288 (33.57)	269 (31.35)	68 (7.92)	3.01 ± 0.86	3 (2–4)
	3. Did you find that the texts were difficult to understand?	18 (2.10)	142 (16.55)	295 (34.38)	330 (38.46)	73 (8.51)	3.19 ± 0.81	3 (3–4)
	4. Did you need a lot of time to understand these materials?	15 (1.75)	104 (12.12)	270 (31.47)	396 (46.15)	73 (8.51)	3.33 ± 0.78	4 (3–4)
	5. Did you need or would you have needed someone to help you understand them?	21 (2.45)	125 (14.57)	244 (28.44)	399 (46.50)	69 (8.04)	3.29 ± 0.83	4 (3–4)
**Subscale**	**Items**	**1—never**	**2—rarely**	**3—sometimes**	**4—often**	** *Missing* **	**Mean ± SD**	**Median (IQR)**
**Interactive/communicative and critical subscale** “*Have you ever thought or been advised to vaccinate yourself against one or more diseases?*”	6. Have you consulted more than one source of information?	60 (6.99)	165 (19.23)	310 (36.13)	262 (30.54)	61 (7.11)	2.97 ± 0.91	3 (2–4)
	7. Did you find the information you were looking for?	20 (2.33)	112 (13.05)	334 (38.93)	327 (38.11)	65 (7.58)	3.22 ± 0.78	3 (3–4)
	8. Did you understand the information found?	18 (2.10)	61 (7.11)	240 (27.97)	464 (54.08)	75 (8.74)	3.47 ± 0.74	4 (3–4)
	9. Have you had the opportunity to use the information?	39 (4.55)	85 (9.91)	340 (39.63)	324 (37.76)	70 (8.15)	3.20 ± 0.82	3 (3–4)
	10. Did you discuss what you understood about vaccinations with your doctor or other people?	78 (9.09)	130 (15.15)	319 (37.18)	267 (31.12)	64 (7.46)	2.98 ± 0.95	3 (2–4)
	11. Did you consider whether the information collected was about your condition?	55 (6.41)	122 (14.22)	297 (34.62)	314 (36.60)	70 (8.15)	3.10 ± 0.91	3 (3–4)
	12. Have you considered the credibility of the sources?	30 (3.49)	82 (9.56)	268 (31.24)	407 (47.44)	71 (8.27)	3.34 ± 0.81	4 (3–4)
	13. Did you check whether the information was correct?	46 (5.36)	85 (9.91)	255 (29.72)	396 (46.15)	76 (8.86)	3.28 ± 0.87	4 (3–4)
	14. Did you find any useful information to make a decision on whether or not to get vaccinated?	47 (5.48)	116 (13.52)	270 (31.47)	351 (40.91)	74 (8.62)	3.18 ± 0.90	3 (3–4)

SD: standard deviation; IQR: interquartile range.

**Table 5 vaccines-10-00682-t005:** Univariate logistic regression analyses: HLVa-IT score (total score, functional subscale score, and interactive/communicative/critical subscale scores) as a predictor of the main sources of information regarding vaccines and vaccinations.

Outcome Variable—Source of Information	Model 1. Covariate: HLVa-IT Total Score *		Model 2. Covariate: HLVa-IT-F Subscale Score *		Model 3. Covariate: HLVa-IT-ICC Subscale Score *	
	OR [95%CI]	*p*	OR [95%CI]	*p*	OR [95%CI]	*p*
Leaflets and brochures	1.25 [0.90; 1.76]	0.190	1.09 [0.88; 1.37]	0.430	1.15 [0.88; 1.50]	0.310
Medical or official posters	0.81 [0.58; 1.13]	0.220	0.76 [0.61; 0.95]	0.020	0.91 [0.70; 1.18]	0.470
TV/Radio	0.70 [0.49–1.02]	0.060	0.75 [0.59; 0.96]	0.020	0.86 [0.65; 1.58]	0.310
Official vaccination campaigns	1.47 [1.05; 2.06]	0.030	0.98 [0.78; 1.22]	0.830	1.49 [1.14; 1.95]	0.003
GP or other health professionals	2.37 [1.66; 3.43]	<0.001	0.82 [0.65; 1.03]	0.090	2.68 [2.01; 3.61]	<0.001
Family members or social networks	0.85 [0.57; 1.25]	0.400	0.79 [0.61; 1.03]	0.080	1.13 [0.83; 1.55]	0.450
Social Media	0.48 [0.27; 0.88]	0.020	1.25 [0.84; 1.90]	0.280	0.46 [0.30; 0.70]	<0.001
Search engines	1.65 [1.15; 2.40]	0.010	0.95 [0.75; 1.21]	0.690	1.63 [1.21; 2.22]	0.002

HLVa-IT-F: HLVa-IT functional; HLVa-IT-ICC: HLVa-IT interactive/communicative/critical; OR: Odds Ratio; CI: Confidence Interval. * Adjusted for educational level, profession, language.

## Data Availability

Data are available for scientific purposes after written request to the corresponding author.

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
