# Peer review of "Vaccine Literacy and Source of Information about Vaccination among Staff of Nursing Homes: A Cross-Sectional Survey Conducted in Tuscany (Italy)"

_vaccines, 2022, doi:10.3390/vaccines10050682_

Round 1
Reviewer 1 Report
In this manuscript entitled ‘Vaccine literacy and source of information about vaccination among staff of nursing homes: a cross-sectional survey conducted in Tuscany (Italy)’, the authors validated the HLVa-IT (Vaccine health literacy for adults in Italian language) for the staff of Nursing Homes, to measure VL and to assess its relationship with the sources used to get information about vaccinations. Because vaccine literacy influence on vaccine acceptance, it is valuable to evaluate VL of vaccine by target population. The manuscript is well designed and investigation is meaningful. For publication, however, the authors should consider several points mentioning below.
Major points:
- Because HLVa-IT is not familiar to reader, including this reviewer, the authors had better describe the items of questionnaire in 2.1 HLV-IT section clearly. Are the score positively or negatively correlate with vaccine literacy ? I was not able to understand the items until I read the results section with Table 4.
- The statistical significance levels were unclear, because they did not show p-values. I am afraid that they did not consider risk of multiple comparisons, thus their results have afraid false positive inference. They compared at least 8 times for souse of information and HL Va-IT subscale for functional subscale and interactive and communicative subscale. If they used Bonferroni correction to avoid false positive interference, the P<0.006 satisfy the statistical significance.
- Because study participants were limited for staff of nursing home, the results were biased from those from general population. They had better discuss this points with referencing previous reports conducted by Biasio et al .
- Because previous reports found the consistent positive correlation for all three VL scales with the educational degree, language and profession might have confounding influence on the association between VL and source of information in this study. Regardin figure 3 and 4, The authors should compare the VL scores with adjusted for language and profession.
Minor points: Power calculation should be mentioned in the method section.
Author Response
Reviewer 1
Because HLVa-IT is not familiar to reader, including this reviewer, the authors had better describe the items of questionnaire in 2.1 HLV-IT section clearly. Are the score positively or negatively correlate with vaccine literacy? I was not able to understand the items until I read the results section with Table 4.
Reply: in the methods section, a detailed description of the HLV-IT has been added.
The statistical significance levels were unclear, because they did not show p-values. I am afraid that they did not consider risk of multiple comparisons, thus their results have afraid false positive inference. They compared at least 8 times for souse of information and HLVa-IT subscale for functional subscale and interactive and communicative subscale. If they used Bonferroni correction to avoid false positive interference, the P<0.006 satisfy the statistical significance.
Reply: Bonferroni correction has been added and p-values reported in the main text.
Because study participants were limited for staff of nursing home, the results were biased from those from general population. They had better discuss this points with referencing previous reports conducted by Biasio et al.
Reply: thanks for this remark. The aim of this study was to validate the HLVa-IT for measuring vaccine literacy among the staff of nursing homes and to measure vaccine literacy in such a peculiar group. This aspect has been clarified in the discussion, when we have compared our results with those of Biasio (general population recruited in the waiting rooms of general practitioners).
Because previous reports found the consistent positive correlation for all three VL scales with the educational degree, language and profession might have confounding influence on the association between VL and source of information in this study. Regarding figure 3 and 4, The authors should compare the VL scores with adjusted for language and profession.
Reply: logistic regression analysis was added in order to taking into account educational level, profession and language while considering the relationship between VL and sources of information. Please, note that figures 3, 4 and 5 were moved into supplementary material and modified, according to the suggestion of reviewer n.2,
Minor points: Power calculation should be mentioned in the method section.
Reply: power estimation has been added
Reviewer 2 Report
Missing data and non response in the study should be reported and discussed.
Authors shall explain in details the difference between analyzed and collected questionnaires (1794 vs 858).
It is essential to explain why these filters considered in the sample selection: "having excluded those workers who have not had previous experiences on written documents on vaccines and vaccinations and those who have ever thought or have been adviced to vaccinate themselves"
There is a conflict in table 4: according to the method part and the header of the table, mean, median and IQRs have been calculated according to the inverse scores.
Figure 3 to 5 can not be informative, because box plots can only show the distribution in subgroups and can not show the differences assessed by mann whitney u test, which assess and compare mean ranks between groups. if sample size in each subgroup is above 30, regardless of the results of KS test, it is recommended to use parametric tests and error bars instead.
One important question remained unanswered in the study, the adequacy of VL in total and subgroups according to a cut off level for questionnaire and its components.
In method, the way questions of the questionnaire selected and the way for measurement of content, construct and criterion validity of the questionnaire have not been discussed. It is essential to explain these validity measures of the questionnaire.
Discussion part shall be revised according to the above corrections.
Decision: Major revision
Author Response
Reviewer 2
Missing data and non response in the study should be reported and discussed.
Reply: Thanks for this suggestion: the lack of discussion of missing data ed excluded questionnaire was a weakness of our paper. We have accepted the invitation and added some statement both in the results and in the discussions.
Authors shall explain in details the difference between analyzed and collected questionnaires (1794 vs 858).
Rely: differences between analyzed and excluded questionnaires were added
It is essential to explain why these filters considered in the sample selection: "having excluded those workers who have not had previous experiences on written documents on vaccines and vaccinations and those who have ever thought or have been adviced to vaccinate themselves"
Reply: thanks for the remark. The filter questions were included in the HLVa-IT developed by Biasio et al, so we have used them also in our study. We have clarified this aspect in the main text.
There is a conflict in table 4: according to the method part and the header of the table, mean, median and IQRs have been calculated according to the inverse scores.
Reply: thanks for this comment. We have modified the header of the table
Figure 3 to 5 can not be informative, because box plots can only show the distribution in subgroups and can not show the differences assessed by mann whitney u test, which assess and compare mean ranks between groups. if sample size in each subgroup is above 30, regardless of the results of KS test, it is recommended to use parametric tests and error bars instead.
Reply: Figures 3-5 were moved to supplementary materials, and error bars were used instead of box plot. Moreover, according to reviewer suggestion, parametric test was used. Please, note that according to the suggestion of another reviewer, Bonferroni correction has been added to avoid false positive interference related to multiple comparisons.
One important question remained unanswered in the study, the adequacy of VL in total and subgroups according to a cut off level for questionnaire and its components.
Reply: the issues of excluded questionnaires and missing values have been discussed. Moreover, power calculation for sample size has been added in the method section.
In method, the way questions of the questionnaire selected and the way for measurement of content, construct and criterion validity of the questionnaire have not been discussed. It is essential to explain these validity measures of the questionnaire.
Replay: in the methods section, a detailed description of the HLV-IT has been added, include more information regarding the original validation study. Please, note that we did not develop a new instrument, but we used an already validated tool to measure VL in a population different from that of the validation study. For this reason, we did not select the items: we considered the entire HLVa-IT (including also the filter questions) and we did not modify it in any sections or items.
Discussion part shall be revised according to the above corrections.
Reply: Discussion has been revised according to the new results.
Reviewer 3 Report
This is an interesting study on a significant topic.
Please consider the following changes to increase the quality of the manuscript:
- Please provide a more informative abstract. The background section is too extensive.
- Please add 2-3 sentences on the vaccine hesitancy/anti-vaccination movement in Italy. This would be interesting for international readers.
- Please provide more precise data on study design and data collection.
- Please specify the online questionnaires used in this study.
- Figures 3-5 may be considered as supplementary materials to focus on the most important data in the main text.
- Please consider 2-3 sentences on practical implications as well as further research needs.
- The limitations section should be more precise and address the limitations of the current methods (direct link).
Author Response
Reviewer 3
Please provide a more informative abstract.
Reply: the abstract has been changed according to the suggestions.
The background section is too extensive. Please add 2-3 sentences on the vaccine hesitancy/anti-vaccination movement in Italy. This would be interesting for international readers.
Reply: in the background, redundant sentences were removed and two sentences on vaccine hesitancy and anti-vaccination movement in Italy - especially with respect to vaccination policy – have been added.
Please provide more precise data on study design and data collection.
Reply: in the method section, more information on study design and data collection have been added
Please specify the online questionnaires used in this study.
Reply: for the purposes investigated in this paper, only the questionnaire filled in by the staff member was considered. This aspect has been better clarified in the methods.
Figures 3-5 may be considered as supplementary materials to focus on the most important data in the main text.
Reply: figures 3-5 have been moved to supplementary materials, and changed according to the comments of the other reviewers
Please consider 2-3 sentences on practical implications as well as further research needs.
Reply: Practical implications have been added at the end of the discussion
The limitations section should be more precise and address the limitations of the current methods (direct link).
Reply: Limitations have been more discussed
Round 2
Reviewer 1 Report
Thank you for following my suggestions, and corrected the manuscript. The authors have improved their manuscript. I don't have further comments.
Author Response
Dear Reviewer,
thanks for you comments.
Kind regards
Reviewer 2 Report
By the modifications performed, the article is acceptable for publication.
Just all p values shall be reported with 3 decimals. in abstract lines 31 and 32 and in the text, Table 6 and lines 355 and 357.
Decision: Accept with minor changes
Author Response
Dear Reviewer,
thanks for your suggestion. P-values have been modified according to your comments.
Kind regards
Reviewer 3 Report
The Authors addressed all the comments.
Author Response
Dear Reviewer,
thanks for your comment.
Kind regards